# Dietary Intakes of Slovenian Adults and Elderly: Design and Results of the National Dietary Study SI.Menu 2017/18

**DOI:** 10.3390/nu14173618

**Published:** 2022-09-01

**Authors:** Matej Gregorič, Hristo Hristov, Urška Blaznik, Barbara Koroušić Seljak, Nataša Delfar, Igor Pravst

**Affiliations:** 1Health Survey and Health Promotion Centre, National Institute of Public Health, Trubarjeva Cesta 2, SI-1000 Ljubljana, Slovenia; 2Nutrition Institute, Tržaška Cesta 40, SI-1000 Ljubljana, Slovenia; 3Computer Systems Department, Jožef Stefan Institute, Jamova Ulica 39, SI-1000 Ljubljana, Slovenia; 4Health Data Centre, National Institute of Public Health, Trubarjeva Cesta 2, SI-1000 Ljubljana, Slovenia; 5Biotechnical Faculty, University of Ljubljana, Jamnikarjeva 101, SI-1000 Ljubljana, Slovenia; 6VIST-Faculty of Applied Sciences, Gerbičeva Cesta 51A, SI-1000 Ljubljana, Slovenia

**Keywords:** dietary survey, dietary intake, energy, macronutrients, 24-h recall, FPQ, adults, elderly, Slovenia

## Abstract

Incomparable and insufficiently detailed information on dietary intakes are common challenges associated with dietary assessment methods. Being a European Union country, Slovenia is expected to conduct national food consumption studies in line with harmonised EU Menu methodology. The present study aimed to describe the methodology and protocols in the Slovenian nationally representative dietary survey SI.Menu 2017/18, and to assess population dietary habits with respect to food consumption and energy and macronutrient intakes. While the study targeted various population groups, this report is focused on adults. A representative sample of participants was randomly selected from the Central Register of Population according to sex, age classes and place of residency, following a two-stage stratified sampling procedure. Information on food consumption was collected with two non-consecutive 24-h dietary recalls using a web-based Open Platform for Clinical Nutrition (OPEN) software. Data were complemented with a food propensity questionnaire to adjust for usual intake distribution. Altogether, 364 adults (18–64 years) and 416 elderlies (65–74 years) were included in the data analyses. Study results highlighted that observed dietary patterns notably differ from food-based dietary guidelines. Typical diets are unbalanced due to high amounts of consumed meat and meat products, foods high in sugar, fat and salt, and low intake of fruits and vegetables and milk and dairy products. Consequently, the energy proportion of carbohydrates, proteins, and to some extent, free sugars and total fats, as well as intake of dietary fibre and total water deviates from the reference values. Age and sex were significantly marked by differences in dietary intakes, with particularly unfavourable trends in adults and men. Study results call for adoption of prevention and public health intervention strategies to improve dietary patterns, taking into account population group differences. In addition, all developed protocols and tools will be useful for further data collection, supporting regular dietary monitoring systems and trend analyses.

## 1. Introduction

Non-communicable diseases are the leading causes of death and disease burden both globally and in Slovenia [1]. Dietary risk factors and physical inactivity are considered as key risk factors for the global disease burden [2]. According to the Global Burden of Disease Study [3], approximately two-thirds of the attributable cause of mortality is associated with dietary risk factors. The diet is mainly linked with cardiovascular diseases, cancer and diabetes, which are also the main causes of morbidity and mortality in Slovenia [4]. Therefore, monitoring and improving dietary patterns in the population is fundamental to ensure adequate nutritional status, and healthy and active ageing [5]. 

Previous nationwide dietary studies conducted in Slovenian adults (in 1995 and 2008) used not fully comparable dietary assessment methodologies and were lacking precision [6,7]. Since food intakes differed substantially from the food-based dietary guidelines and nutrient intakes deviated from the national reference values, new detailed data are needed for the development and implementation of nutrition and food safety policies, and to evaluate efficiency in the existing Slovenian National Food and Nutrition Action Plan. In this context, detailed collection of food consumption data at individual level is essential [8].

A variety of methods is available for the collection of food intake information, and suitable for population-based studies in the adult population. Different methods have different advantages and disadvantages [9]. It is well acknowledged that the misreporting or under-reporting of food intake are key limitations in all dietary assessment studies. Research on this has been affected by a lack of consensus on the assessment methodologies [10]. A major step forward has been achieved with support from the European Food Safety Authority (EFSA), which followed a large-scale validation project [11] and harmonised a European consensus, that two non-consecutive 24-h dietary recalls (24HDR) are the most appropriate method to collect food consumption data in adults [8]. Furthermore, in recent years, several computer-assisted dietary intake assessment tools (such as software, web-based, or mobile applications) have been developed to facilitate and simplify the recording of food consumption, assuring a higher level of accuracy and enabling more efficient data analyses [9]. 

Regardless of the used method, self-reported food intake relies on the participant’s memory and underestimates true food and nutrient intake [12]. Since the accuracy of dietary intake assessment is important in dietary studies, the data collected by the 24HDR depend on the accuracy of the reported data on the type of consumed foods, precision of the description, the recipes, and precision of the consumption quantities. Since the respondents have limited ability to precisely quantify the amount of consumed food, different portion size measurement aids are recommended. To this end, the EFSA has advised countries to develop their own country-specific and validated picture books to support quantification of food consumption in the recalls [8]. 

Since 24HDR is considered as a method of choice to collect short-term data in dietary assessment studies, scientists have developed several different statistical methods to model ‘normal’ and usual nutrient intake. Such methods notably minimise within-person variability of short-term measurements on a small number of days, caused by the day-to-day variation in diet and random errors. Use of such methods enables better estimation of the true distribution of usual intakes, that would be observed on longer-term [13]. 

A key objective of the EFSA was to conduct a pan-European survey with harmonised instruments and protocols, supporting the compilation of comparable food consumption datasets across the European Union (EU) member states. This is crucial also for policy makers on the EU level, because such data enable dietary exposure assessment for the purposes of risk assessment and assessment of the nutrient intakes, which is also needed for setting appropriate dietary reference values for specific population (sub)groups [14]. Therefore, Slovenia has decided to join this initiative and conduct a third Slovenian representative dietary survey, named SI.Menu 2017/18. 

The aim of the study presented in this paper was to assess dietary habits with respect to food consumption and energy and macronutrient intakes among different age groups in the Slovenian population and to describe the design, methods. and protocols used in the Slovenian national dietary survey SI.Menu 2017/18.

## 2. Materials and Methods

### 2.1. Design and Study Sample

The study design was based on the EFSA Guidance on EU Menu methodology [8] and the recommendations made after the pilot study in the view of a Pan-European dietary study—Adolescents, adults and elderly (PILOTPANEU) [15]. Within the SI.Menu project, we adapted available methodology for use in the national cross-sectional food consumption survey, according to EFSA Guidance on EU Menu methodology, conducted data collection, and data transfer to EFSA using harmonised formatting and food categorisation. The study was carried out by a consortium consisting of Slovenian National Institute of Public Health (as a project leader) and seven other research organisations (see Acknowledgement section), and supported by the EFSA and Ministry of Health of the Republic of Slovenia. The study was conducted in accordance with the Declaration of Helsinki. The study protocol was approved by the National Medical Ethics Committee (NMEC) of the Republic of Slovenia (no. KME 0120-337/2016); participation was voluntary and informed consent was obtained from all enrolled subjects.

The detailed study design and sample are described in detail elsewhere [16]. In short, the subjects of the SI.Menu survey were selected according to a two-stage, stratified sampling procedure. Sampling was stratified by size and type of settlements, place of residency (to cover all NUTS-3 statistical regions), sex, and age groups. Individuals were randomly selected from Central Register of Population (CRP) of Slovenia by the National Statistical Office. Sampling was carried out for specific individuals, meaning that selected persons could not be substituted with another household member.

After systematically excluding subjects (i.e., institutionalised persons, living abroad), invitation letters were sent to selected individuals’ home addresses in Slovenia. A sampling frame included 2280 potential subjects aged 10–74, according to the EFSA recommended age groups. Invited subjects were classified into three age groups—adolescents (10–17 years), adults (18–64 years), and elderlies (65–74 years). In total, 1319 subjects met inclusion criteria and had fully completed the survey with two 24-h recalls, which was conducted from March 2017 to February 2018. The data collection period was divided into four quarters (3-monthly samples) with equal distribution of week and weekend days, in order to incorporate seasonal effects and day-to-day variability in food consumption. Altogether, the response rate was 62%. For the purpose of this report, data analysis was conducted only on eligible adults (18–64 years old; N = 364) and elderlies (65–74 years old; N = 416).

### 2.2. Data Collection

Data were collected during two computer-assisted personal interviews. A general questionnaire was used to obtain information on socio-demographic and socio-economic, as well as health related outcomes and lifestyle factors, including consumption and eating habits, and food allergies. Subjects were also asked to provide self-reported usual frequency and duration of physical activity, which were later converted to the International Physical Activity Questionnaire (IPAQ) score using previously described method [17]. Food intake was assessed with two non-consecutive 24-h recalls (HDR). This standard dietary assessment method was complemented with the Food Propensity Questionnaire (FPQ), which was chosen to collect data on respondents’ usual frequency of consumption of specific foods (and dietary supplements) in the last 12 months, enabling further modelling of usual intakes [8,18]. In total, 75 food items corresponding to nine food groups were used. 

The first face-to-face interview took place at the respondent’s home address and was consisted of explanation of study procedures and administration of general questionnaire, FPQ, first 24HDR, as well as anthropometric measurements. The second interview was performed either over the telephone, or at participants’ homes, and consisted of a second 24HDR and controlling collected data questionnaire. Between the first and second interviews, there was a minimum seven day and maximum three week interval. 

Body mass and height were measured using standardised medical equipment and procedures. Body mass index (BMI) was calculated by dividing body mass by the square of their height in meters of participants and assessed using BMI cut-off points of <18.5, <25.0 and ≥25.0 kg/m^2^ for underweight, normal weight, and overweight, respectively [19]. Primary interest in measuring body mass and BMI in our case was to assess under- and over-reporting.

### 2.3. Collection of Food Consumption Data

The National Institute of Public Health professionals in human nutrition and interviewing techniques performed two-day training for the researchers who conducted the interviews. Each researcher received methodological instructions and materials. Data collection was supported by web-based OPEN application [20,21], and with a nationally adjusted and validated picture book [22], enabling more precise estimation of the amounts of consumed foods. The trained researchers performed face-to-face interviews with each of the respondents on two non-consecutive days. The recalls were distributed over all days of the week and seasons. Researchers collected detailed descriptions of the type and amount of food consumed on the previous day, also specifically asking for key food items, that could be easily forgotten (salt, water, oil, etc.) [8]. For home-cooked (composite) dishes, respondents were asked to provide recipes. When this was not available, standard recipes from the OPEN app. [23] were used. All composite dishes were disaggregated, providing information on the ingredient level.

Amounts of consumed food items were estimated using a national picture book, complemented with household measures and portions of typical dishes. The picture book was composed of 46 different food products or simple dishes, and presented with a series of pictures of 6 different portion sizes [22]. For the purposes of this study, reported food items were classified into 21 main food groups, and additional sublevels, according to the modified categorisation system adapted from Haubrock et al. [24] and corresponding to the food-based dietary guidelines (FBDG) based on DGE Nutrition Circle [25], with consideration of food categories in the FPQ. 

### 2.4. OPEN Software

Food consumption data were collected using a mobile application based on the Open Platform for Clinical Nutrition (OPKP developed by Jožef Stefan Institute, Ljubljana, Slovenia) [23]. In short, OPEN is a national platform, which was developed for planning and assessment of dietary intakes, both in research and clinical practice. Within the SI.Menu project, OPEN was upgraded to support the 24-h recall/food diary method in line with the EFSA Guidance [8]. It allows information about food and beverages consumed during the survey days to be entered (multi-pass method). OPEN is linked with a food composition database, enabling conversion of reported amounts of consumed food/drink into energy in nutrient intakes data. To calculate composition data for traditional and frequently consumed Slovenian dishes, OPEN applies a recipe-calculation procedure, originally recommended by the INFOODS and recommended by the EuroFIR [26]. OPEN includes the Slovenian food composition database [21], which is complemented with international food composition datasets [27]. The use of OPEN has been described in detail elsewhere [20].

All foods and beverages reported in 24HDR were inserted to OPEN, and linked with food composition data. For each complex dish, a recipe with preparation procedure was selected, enabling automatic calculation of the quantities of specific food ingredients. OPEN also enabled insertion of new recipes, where appropriate. 

### 2.5. Under- and Over-Reporting and Additional Exclusion Criteria

Handling under- and over-reporters was previously described and explained [28]. In short, estimated subjects’ energy intakes were assessed using the cut-off points method initially described by Goldberg et al. [29] and further adapted by Black et al. [30]. The method is based on the ratio of reported daily energy intake and basic metabolic rate (BMR). The calculated cut-off points for 24HDR for under- and over-reporting were 0.41 and 2.46, respectively, resulting in exclusion of 36 subjects. Furthermore, subjects reporting energy intakes of less than 500 kcal, those with incomplete anthropometric data, or/and 24HDR data, or/and general questionnaire/FPQ, were also excluded from further analyses. Altogether, data from 72 individuals (30 adults in 42 elderlies) were excluded from further analyses.

### 2.6. Data Analyses

To calculate usual daily dietary intake of foods and nutrients, two 24-h recalls and FPQs were used. Day-to-day inter- and intra-individual variations in the intake distribution was modelled using the Multiple Source Method (MSM) [24], using age, sex, and BMI as covariates. The MSM modelling approach used FPQ data to correct for within-individual variation in food and nutrient intake, providing data on usual dietary intake on an individual level [31]. To avoid a problem of high level of non-consumers in the modelling of usual food intakes for individual food items, MSM was used on larger food groups. We used food (sub)groups, adapted from Haubrock et al. [24]. It should be noted that for eggs, data are reported only as 24HDR intake, because this food group was not included into FPQ. Modelling of usual daily food intakes with MSM was carried out with transformed FPQ frequencies (i.e., FPQ frequency 1 time/week was converted to 1/7 per day). Food groups were subsequently used to also estimate energy and nutrient intakes on the individual level and thus using a combination of 75 low-level FPQ and 36 upper-level literature adjusted food category groups [25]. 

The descriptive data are presented as mean and standard deviation for the continuous variables or as numbers and percentages for categorised variables. Usual intake data (food groups, energy, nutrients)—calculated separately for men/women and adults/elderlies, are presented as mean (per day) with standard deviation, and with median/percentiles (5, 25, 50, 75, and 95%). Nutrient intakes are calculated both in g per day, and as percentage of total energy intake (TEI). For energy and proteins, intake is also presented in kJ or g per kg body weight, respectively. Individual intakes for total/free sugars and dietary fibre are taken from previous reports ([28] and [32], respectively).

Energy and nutrient intakes were compared with D-A-CH reference values [33], which have been adopted by the Ministry of Health of the Republic of Slovenia. Total water intakes were compared with adequate intake in EFSA’s dietary reference values for water [34]. We calculated the proportion of subjects (%) not meeting sex and age-group adapted reference values. The same approach was used for food groups, where comparison was carried out using cut-offs of FBDG, originating from DGE Nutrition Circle [25], which are based on the D-A-CH reference values. Where DGE FBDG are refereeing to weekly amounts (i.e., for meats and eggs), comparison was carried out with cut-off values, converted into daily amounts. 

All statistical analyses were performed using STATA (version 17.0; StataCorp LLC, Colledge Station, TX, USA).

## 3. Results

Analyses were performed on adult and elderly participants of the Slovenian national SI.Menu 2017/18 dietary survey. Using the quota sampling approach, we were able to gain similar sample sizes for adults aged 18–64 years (46.7%; N = 364) and elderlies aged 65–74 years (53.3%; N = 416). Sample characteristics are described in Table 1. In short, a higher proportion of participants were categorised in the rural living area group, lower education level group, overweight/obese BMI groups, employed group, and in east the cohesion region group. A mean BMI (mean ± SD) of the participants in our study was 26.7 ± 5.2 for adults and 28.4 ± 5.0 for elderlies, respectively. Altogether, 39.6% of adults and 24.8% elderlies were normal weight (BMI < 25.0), while 59.3% and 74.0% were overweight/obese (BMI ≥ 25.0), respectively. 

### 3.1. Food Consumption per Food Groups

SI.Menu study results, presented in Table 2 and Table 3, revealed that for both males and females, milk and dairy products were generally consumed in higher amounts among the adults compared to the elderlies. For milk, sex differences were more notable among the elderly, where more milk (mean ± SD) was consumed by females (76.3 ± 80.8 g) compared to males (59.9 ± 83.6 g), while a minor difference was noted among adults. However, 13.5% of adults were declared as non-consumers of milk. A very similar pattern was observed for dairy products, where more extensive sex differences were again observed among the elderly; consumption of dairy products (mean ± SD) was 129.4 ± 123.5 g and 91.2 ± 102.5 g for females and males, respectively. 

The results show that adults in general consume similar amounts of vegetables as the elderly. The highest amounts (mean ± SD) were consumed by elderly females (163.4 ± 74.2 g)—compared to elderly males (160.1 ± 86.2 g), while a contrary trend was observed in adults (158.3 ± 77.1 g for females, and 163.0 ± 64.8 g for males). Within the vegetable food groups, males consumed higher amounts of preserved and canned vegetables in both age groups. 

Mean daily fruit consumption was higher, compared to vegetables. The results show that the elderly consumed more fruit per day than the adults. In both age groups, females reported a higher amount of consumed fruit than males. Among the adults, females consumed (mean ± SD) notably more fruit (226.0 ± 151.3 g) compared to males (162.2 ± 142.5 g), while in the elderlies, this difference was minor (267.8 ± 155.8 g and 210.7 ± 163.8 g, respectively). Preserved and dried fruits were consumed in higher amounts particularly in the elderlies, where higher amounts were observed for females (38.8 ± 87.4 g) and lower for males (30.4 ± 82.5 g).

Adults generally reported a higher consumption of meat and meat products than the elderlies. In both population groups, notably higher consumption was observed in males. In adults, mean daily consumption quantities (mean ± SD) were 208.6 ± 101.8 g for males, and 140.8 ± 76.2 g for females. A similar but less expressed trend was observed in the elderlies, where consumption of meat and meat products were 188.7 ± 88.6 g and 141.1 ± 80.6 g, respectively. Only 1.6% of adults were declared as non-consumers of meat. Looking to a sub-group of meat products, we observed almost double consumption in males, in comparison to females. In adults, consumption of meat products was 51.5 ± 62.6 g in males and 27.7 ± 43.4 g in males. A similar trend was observed in the elderlies.

The data related to consumption of fish and fish products show higher amounts for adults, compared to the elderly population. Among the adults, notably higher amounts (mean ± SD) were again observed for males (26.5 ± 58.0 g), compared to females (18.2 ± 46.7 g). A contrary consumption trend was observed in the elderlies (21.2 ± 50.1 g for females and 19.8 ± 47.9 g for males). About half the amount of this food group was consumed as a canned product, where sex differences were even more extensive. 

The analysis of consumption of eggs shows a higher consumption amounts in adults, in comparison to the elderlies. In both age groups, males consumed more eggs compared to the females. Using 24HDR data, almost half of elderlies did not report consumption of eggs.

Legumes, a possible meat substitute, were consumed in higher amounts among the elderlies, compared to adults. In both age groups, more legumes were consumed in males compared to females. In the elderlies, daily intakes were 21.3 ± 40.3 g in males and 17.4 ± 30.7 g in females, while this was 14.6 ± 32.0 g and 11.6 ± 23.8 g in adults, respectively. 

Among starchy foods, bread and bakery products presented about half of the total intake. Higher consumption amounts (mean ± SD) of bread and bakery products were observed in males-particularly in the adult population group (210.0 ± 96.8 g for males and 144.7 ± 81.0 g for females). A similar but less expressed pattern was observed in the elderlies. The highest amounts (mean ± SD) of bread were consumed by elderly males (178.1 ± 76.1 g). Among cereal products, the highest intake of pasta and rice was observed for adult males (62.8 ± 57.8 g), and the lowest intake was observed for elderly females (10.4 ± 22.4 g). In both age groups, we observed a higher consumption of breakfast cereals in females. It should be noted that about half of the elderlies were non-consumers of breakfast cereals, while in adults this was about a quarter. One of the major sub-groups in this category was also potato; in both age groups, higher intakes were observed in males (99.0 ± 70.70 g for adults and 99.0 ± 64.3 g for elderlies), in comparison to females (75.7 ± 61.2 g for adults and 87.8 ± 50.8 g for elderlies). 

Consumption of fats and oils was in general higher in the elderlies, in comparison with adults. The ratio between consumed vegetable and animal fats is about 3:2 both males and females—in both age groups. In adults, intake (mean ± SD) of fats and oils was higher in males (28.2 ± 20.5 g) compared to females (22.6 ± 14.0 g), while comparable consumption was observed among the elderlies.

A higher consumption of high sugar foods was observed in adults, particularly in females. Among the adults, intake (mean ± SD) was 106.5 ± 104.6 g in females and 98.2 ± 121.1 g in males, while a less notable difference was observed among the elderlies. Sweet desserts presented over one-third of this food group, again with the most notable sex difference in adults (65.7 ± 93.7 g in females and 46.4 ± 94.0 g in males). 

Ready to eat meal intakes in males and females were quite comparable, in both adults and the elderlies. Consumption in these groups varied between 26.3 g (for elderly males) and 29.2 g (in adult males). 

Adults generally consumed more fluids (water, drinks, beverages, etc.) compared to the elderlies. Sex difference was particularly notable for sugar-containing soft beverages, where in adults the intake (mean ± SD) was 136.4. ± 253.4 mL for males, and 62.6 ± 177.2 mL for females. Notably lower consumption quantities were observed in the elderlies (40.7 ± 128.7 mL in males and 24.0 ± 91.9 mL in females), who also consumed less fruit/vegetable juices, but more hot drinks. It should be noted that more than a half of the elderlies were declared as non-consumers of sugar-contained soft beverages, and one-third as non-consumers of fruit and vegetable juices. In both age groups, more tap/bottled water and hot drinks were consumed by females, in comparison to males. This pattern was observed particularly in adults, with hot drink intakes of 323.1 ± 267.1 mL in females and 243.7 ± 207.2 mL in males. 

Adults generally consumed more alcoholic beverages than the elderlies. In both age groups, males drank notably more alcoholic beverages than females. Looking into specific alcoholic drinks, in males we observed the highest consumption of beer—183.3 ± 314.9 mL in adults, and 112.9 ± 234.2 mL in elderlies. Quite different situation was observed in females, which consumed much less beer (44.0 ± 145.6 mL and 8.4 ± 48.4 mL, respectively). The second most consumed alcoholic beverage was wine. The highest consumption of wine was observed in males—32.8 ± 62.8 mL in adults, and 78.5 ± 135.6 mL in elderlies.

### 3.2. Intakes of Energy, Nutrients, Dietary Fibre and Water

Study results, presented in Table 4 and Table 5, revealed notably higher daily energy intakes (after excluding under/over-reporters) in adults than the elderlies, as well as in males, when compared to females. Sex difference was more pronounced among adults, where daily energy intake (mean ± SD) was 9587 ± 2663 kJ in males, and 7356 ± 1749 kJ in females. Somewhat lower, but still notable sex differences were also observed among elderlies.

Adults generally reported a higher total water intake than elderlies, as well as more water was consumed by males compared to females in both age groups. For example, mean total water intake (from all sources) in adults was 2438 ± 691 mL for males, and 2260 ± 649 mL for females. In adults, the proportion of energy (mean ± SD) derived from carbohydrates was notably higher among females (45.6 ± 8.4%) compared to males (41.9 ± 8.2%), while this sex difference was less expressed in elderlies. Within carbohydrates, a higher intake (mean ± SD) of total sugars was observed among adult females (15.8 ± 6.0 g) compared to males (12.3 ± 5.6 g), where sugar intake presented about one-third of the energy derived from carbohydrates. Differences between the elderlies and adults were not observed. The proportion of energy (mean ± SD) derived from free sugars in adults was minimally higher in females (7.1 ± 5.1%) compared to males (6.7 ± 5.4%), while some difference was also observed in the elderlies (5.5 ± 4.1 and 6.7 ± 4.7%, respectively). In adult males and females, intake of dietary fibre was slightly higher (21.4 ± 7.5 and 20.3 ± 6.6 g, respectively) compared to the elderly counterparts (21.5 ± 6.9 and 21.6 ± 7.0 g, respectively), without major differences between both sexes.

Similarly, adult males and females consumed a slightly higher proportion of energy derived from fats (26.6 ± 5.5 and 26.6 ± 5.6%, respectively) compared to the elderlies (25.7 ± 5.9 and 25.7 ± 5.7%, respectively), without major sex differences. 

The mean proportion of energy derived from proteins was slightly higher among males compared to females, and among adults compared to elderlies. In terms of intake of proteins per kg of body weight (mean ± SD), adult males and females consumed more proteins (1.36 ± 0.48 and 1.20 ± 0.41 g, respectively) than their elderly counterparts (1.25 ± 0.42 and 1.13 ± 0.39 g, respectively).

### 3.3. Components Comparison of Estimated Usual Intakes with Dietary Recommendations

Taking a closer look at those who did not meet the selected DGE food-based dietary recommendations [25], we found that most adults and elderlies were consuming inadequate quantities of recommended food groups, such us milk and dairy product and vegetables. Only 14.2% of adult males and 17.2% of females, and 11.3% elderly males and 9.6% females consumed recommended quantities of milk and dairy products. On the other hand, only 3.5% adult males and 6.8% females, and 9.4% elderly males and 8.9% females consumed enough vegetables (fresh and preserved). For fruits, the lowest consumption was observed among adult males; only 33.1% of them consumed sufficient quantities. Water and non-alcoholic beverages were also consumed in inadequate quantities, especially in the elderly group, where only 11.8% males and 12.3% females reached recommended quantities. On the contrary, in case of (fresh and processed) meat, most males (97.7% of adults and 97.2% of elderlies) exceeded the upper limit of recommended quantities. 

Results presented in Table 6 also revealed that most adults and elderlies had insufficient intake of some macronutrients according to the D-A-CH dietary reference values [33]; this was particularly notable for carbohydrates, but also for dietary fibres. For the proportion of energy derived from carbohydrates, the lowest intake was observed among males, where only 14.5% adults and 16.0% elderlies reached the lower limit of the reference values. Results also revealed insufficient intake of dietary fibres; only 13.3% of adult males and 8.4% of females, and 10.8% of elderly males and 13.3% of females reached the lower limit of reference values. 

For total water, comparison was made with EFSA’s adequate intake [34], which was set separately for males (2.5 L) and females (2 L). In both sex groups we observed high prevalence of insufficient total water intake; this was more expressed in males (58.4% in adults and 72.3% in elderlies) than in females (37.2 and 48.3%, respectively). 

On the contrary, absolute daily intake of proteins was better covered, particularly in adults. Most adults (97.1% of males and 91.6% of females) as well as elderlies (92.5% of males and 86.7% of females) met reference for total daily protein requirements, which are set separately for males/females. However, quite different situation is observed when looking to grams of protein per kilogram of body weight, where reference values considerably differ between adults (>0.8 g/kg b.w.) and elderlies (>1.0 g/kg b.w.). Using these criteria, protein intake was still sufficient in most adults (89.0% in males and 82.2% in females), but not in the elderlies (only 29.1% and 43.3%, respectively). 

## 4. Discussion

This study provides the most recent overview of information on food consumption as well as energy, macronutrients, dietary fibre, and water intakes in a nationally representative sample of adults and elderly in Slovenia, as well as compliance with dietary recommendations. 

### 4.1. Consumption of Different Food Groups among Age and Sex Groups and Comparison with Dietary Recommendations

The study results revealed that food consumption in adults and elderly population commonly differ from food-based dietary guidelines. Age and sex are significantly marked by differences in quantity consumption of individual food groups, which is quite understandable due to different dietary requirements of these population groups. In this respect, adults compared to elderlies and males compared to females stand out the most, as they consume the notably more meat and meat products. High consumption of meat and meat products in amount of more than 1.4 kg per week, which was observed in adult males, could results in health risks [36]. It is particularly concerning that the highest consumption (P95) among adult males corresponds to 3.8 kg of meat and meat products per week. Only 1.6% of adults were declared as non-consumers of meat, which is in line with the data of those who defined themselves as having a vegan or vegetarian diet, and also comparable to the data of other European and neighbouring countries [37]. On the contrary, fish and fish products were consumed in insufficient quantities; slightly more than one-quarter of adults and about one-fifth of elderlies do not reach the lower reference values [25]. Sufficient consumption of fish (instead of meat products) is beneficial both for balanced dietary intake of fatty acids (less saturated fatty acids and more unsaturated and essential fatty acids such as n-3 LCPUFA), as well as for intake of quality protein, vitamin D, selenium, and iodine [38].

Milk and dairy products are consumed in insufficient quantities, which ranks Slovenia among the EU countries with the lowest consumption rates. About three-quarters of adults and the elderlies did not meet the lower reference value [25]; this was particularly pronounced among males compared to females. In total, 12.5% of adults were declared as non-consumers of milk, which is also in line with the declining trend in its consumption among adults in the last decade [39].

It is recommended that at least 250 g of fruit and 250 g of vegetables be consumed daily, due to the content of vitamins, minerals, dietary fibre, and other protective substances [25]. In the long term, this protects against various chronic diseases, including obesity. According to the quantitative recommendations of a healthy diet, there is a lack of vegetable consumption since the majority of study population do not meet the reference values [25]. Slightly higher amounts of vegetables were consumed by the elderlies compared to the adults, for whom it is still unpopular or less accessible. However, it is encouraging that in recent years, after a long period of decline, the consumption of vegetables among adults has started to increase [39]. Furthermore, legumes were also consumed in low amounts. It should be noted that legumes are an important source of plant protein and a possible replacement for meat, and therefore an unexploited opportunity for sustainable diets and food strategies [40].

Foods high in sugar, fat, and salt (e.g., high sugar foods, ready to eat snacks, sugar-containing beverages, and processed meat) are less recommended and therefore intended for occasional use in smaller quantities [25]. These foods should be presented in a very small proportion of our diet, while in our study they represent as much as a quarter of the total daily amount of food consumed. Notably more of these non-recommended foods are consumed by adults compared to the elderlies, and among males. 

Compared to females, males consume more cereals/cereal products, potatoes, oils and fats, and particularly meat and meat products. Females consume more vegetables and fruits, and more milk and dairy products, as well as tap/bottled water, which is in line with the findings that females typically care more about their health than males [41]. There are also notable differences between both age groups; adults consume more potatoes, meat and meat products, where limited consumption is recommended, while elderlies consume more cereals and cereal products, as well as fruit and vegetable. Several studies have concluded that elderlies care more about their health and better follow dietary recommendations in comparison to younger groups, which are less affected by chronic diseases and other health problems [42]. Reported findings are also consistent with previous data for the Slovenian adult population [6,7].

### 4.2. Energy, Macronutrients, Dietary Fibre, and Water Intakes among Age and Sex Groups and Comparison with Dietary Recommendations

The study results also revealed that the average diet of adult and elderly populations mostly does not differ from dietary guidelines for energy intake. However, it should be noted that we used a low physical activity level (PAL 1.4), that describes sedentary lifestyle with moderate activity, as a general reference for total energy expenditure [43]. According to our findings, a certain proportion of the population has moderate or even high physical activity levels, and energy intake might be inadequate for these subgroups of individuals. While our results are comparable with data reported for Swedish, German, and Portuguese adults and/or elderlies [44,45,46], detailed comparisons between studies are difficult because of different dietary assessment methods and differences in sampling approaches and resulting composition of study samples [13]. Furthermore, self-reported instruments, such as IPAQ used in our study, could overestimate PAL in population-based studies and their validity is therefore limited [47]. 

When comparing the percentage of energy provided by the three main macronutrients with the reference values [25], mean proportions of energy derived from proteins and carbohydrates mostly deviated from recommended values. These findings are very similar to dietary intake patterns observed a decade ago among Slovenian adults [6]. Similar to our observations, some dietary studies [44,45,46] also revealed that the proportion of energy from proteins is mostly sufficient, while proportion of energy from carbohydrates was lower than recommended. This discrepancy was particularly pronounced among adults compared to elderlies, and in males compared to females. But we should note that elderlies, where higher intake of protein is recommended (at least 1.0 g/kg body weight) [48], are often at risk of protein deficiency. The majority of elderlies in our samples (70.9% males and 56.7% females) did not meet this recommendation, which is concerning, because sufficient protein intake is crucial in this population group for optimal preservation of lean body mass and body functions [49]. Furthermore, physical activity with sufficient protein intake is also important to reduce risk for sarcopenia [48,49]. 

In both adults and elderlies, the majority of our sample met recommendations that intake of total fats should be below 30% of total energy intake. Nevertheless, around one-quarter of the study population exceeded recommended total fat intake; this was more notable in adults and males. Similar fat intake patterns and related health concerns were found among the adults and elderlies in other countries [44,45,46]. It should be noted that fats are a very complex nutrient, composed of less (i.e., saturated, trans fatty acids) or more desirable (i.e., various unsaturated, including omega-3 fatty acids) constituents, with different intake recommendations. While the SI.Menu dataset has been previously used to estimate dietary intakes of most concerning trans fatty acids—which was found low [50], we are also planning a study, which will provide further insights about intake of other types of fats. However, a major obstacle for such studies is limited data about the fatty acid composition in national food composition database, which should be first supplemented.

Several public health organisations have recommended limiting intake of free sugars to less than 10% of TEI, and preferably below 5% for additional health benefits [35]. While intake of sugars in the Slovenian population has already been described in detail [28], herein we also reported intake of free sugars, as this enabled more meaningful overall comparison of different goals of dietary recommendations. Our results indicated that intake of free sugars is a more notable concern among adults, compared to elderlies, and among females, compared to males. The majority of the population did not exceed the limit of 10% of TEI from free sugars, and about one-quarter of the study population exceeded such recommendation. It should be noted that total consumption of added sugars was a major concern a decade ago; observed improvement in this area might be explained both by changes in dietary behaviours (i.e., reduced consumption of sugar-containing foods and beverages), and in the food supply (i.e., reduced sugar content in available foods) [28].

Intake of dietary fibre has also been previously described in detail [32], and is reported herein for completeness of the comparison—expressed also in grams of dietary fibre per MJ of energy intake. The majority of adults (86.7% males, and 91.6% females) as well as elderlies (89.2% and 86.7%, respectively) did not meet recommended daily intake of 30 g of dietary fibre intake. A similarly low dietary fibre intake was observed in previous national studies, where the lowest dietary fibre intake was observed in overweight/obese subjects [32]. Quite low intakes of total water were also observed in our study. This observation is particularly concerning for elderlies, who are commonly not sufficiently hydrated [51]. Among the elderlies, 48.3% females and 72.3% males reported insufficient total water intake, which might explain results of a previous study on Slovenian elderlies, where about half of the population was clinically dehydrated [52].

### 4.3. The Suitability of Used Methodologies for Assessment of Dietary Intakes in a Population Study

Dietary intake assessment tools have been tested for the research applications on adult or elderly populations. Web-administrated 24HDR protocol on two non-consecutive days using OPEN dietary application was found as feasible method for assessment of actual dietary intakes in both adults and elderlies. Nevertheless, some authors highlighted that 24HDR is less sensitive and a far more accurate dietary assessment of energy and nutrient intake, and suggested use of several days’ dietary records [9]. 

Limited usability of two 24HDR method for estimating usual (long-term) intakes has been addressed with statistical modelling, where FPQ data are also used. In addition, conduction of 24HDR was supported by the validated national picture book, which assisted estimation of portion sizes. As with any population measure, information on dietary intakes is also subject to random and systematic errors. However, many researchers who validated different tools for assessment of self-reported food intake, highlighted that measurement errors depend on several factors, that cannot be sufficiently corrected—such as memory capacity of the subjects, cooperation, and communication skills at one side, and their perception and ability to report portion sizes, at the other side [9,53]. It was observed that insufficient training of researchers and insufficient checking of the reported data can notably affect the accuracy of the data collected [54]. Therefore [9], all researchers who collected recall data in SI.Menu survey underwent a two-day practical educational workshop on dietary surveillance approaches. Aside from the initial training, researchers were also encouraged to contact one of the project leaders should a question or problem occur during the interview.

In general, evaluation of the validity of self-reported methods for estimating daily energy intake provides a valuable check on the general quality of the food intake data, particularly among overweight and obese individuals, where under-reporting is more common [55]. Nevertheless, a common challenge in all dietary intake studies is handling under- and miss-reporting, and different interviewing techniques are used to minimise such errors [14]. Our results revealed that less than 2% of individuals were excluded from further analyses due to under/over-reporting, which is comparable or even less than in many similar dietary studies. However, it should be noted, that a large proportion of our study participants were overweight or obese, which might have a greater tendency to underreport their diary intake [56]. On the other hand, these high rates of overweight and obese were comparable with the national data and did not deviate from the national average [57]. This issue should be further explored in the future.

### 4.4. Strengths and Limitations

In addition to valuable results, this study also showed that the used dietary assessment methods and tools are feasible for use in adults and elderlies, supporting conduction of more regular monitoring in the future. The key strengths of this study are the nationally representative sample with age and sex quotas, as well as detailed food consumption data, collected at the brand level. Secondly, long-term usual intakes were estimated using both 24HDR and FPQ data with statistical modelling, to adjust for intra-individual variability; such procedure allowed better estimate of the true distribution of usual intakes and removal of extreme unlikely values [9,24]. While 24HDR is a standard method for dietary assessment, we should mention that this approach can result in misreporting or under-/over-reporting. While these issues are a key limitation in all such studies, we minimise those by a high-level of researcher-participant interaction (two-visits by trained researchers within 3 weeks) and resulted in low level of incompliance, measured as under- and/or over-reporting of energy intakes. Additional methods for increasing participation rates and compliance were the motivation and surveillance of researchers as well as awarding participants with small incentives. To avoid socio-economic bias in the sample, non-financial compensations were provided to participants. We should also mention that data collection was supported by very useful aids, such as a picture book and a list of commonly forgotten foods in the 24HDR. 

On the other hand, it is important to mention some technological and methodological limitations. In certain conditions, slower speed of OPEN software impacted matching/identification of foods reported during the 24HDR interview, and might have affected identification precision of reported food items. The food items browser is open-ended and allows different approaches of identifying individual food items, which can result in slight differences in results, based on differences in the composition of the selected food item. It was previously highlighted [58] that web-based recalls are very sensitive to the reported name of the food item and spelling errors. If there is no food composition data that fully matches with reported food item, a set of closely matched food items is provided, but some of these might not be an optimal choice [59]. Therefore, we cannot completely exclude some errors in the food coding and in the nutritional composition of reported foods [54]. Furthermore, the abundance and the variability of preparations of the same national recipe can cause errors, since different individual ingredients and preparation procedures require specific corrections in calculations [60]. It should also be noted that although we had a rather high response rate in our study, responsiveness is a major challenge in nutrition studies and will also need to be appropriately addressed in future research. There is increasing evidence that particularly young individuals, male participants, those with poor health, and those with lower socio-economic status are often not able and not willing to participate and need more motivation for participation [8]. 

## 5. Conclusions

To summarise, the study results revealed that the diet of adults and elderlies in Slovenia notably differ from food-based dietary guidelines. Major issues are high intakes of meat and meat products, foods high in sugar, fat and salt, and low intakes of vegetables and fruits, and milk and dairy products. Estimates of energy and nutrient intakes highlighted challenges for carbohydrates, proteins, to some extent in free sugars and total fat intakes, as well as for dietary fibre and total water. Notable differences in the intake of foods and nutrients were observed between both age groups and sexes. Study results support the development of evidence-based programmes and policy measures for supporting healthier food choices, taking into account specifics of different population groups, such as fiscal policies to limit access to unhealthy choices, restricting the marketing of unhealthy food, reformulation of food and meals, improving the quality of meals in institutions, and nutritional education in all periods of life. These measures should be more specifically addressed in the next Slovenian National Food and Nutrition Action Plan. The study results can also serve as a scientific background for the preparation of country-specific food-based dietary guidelines, which Slovenia has not developed, and can also serve as an important measure in this area.

In addition, we showed that the used methodological approach—web-administrated 24HDR combined with FPQ, is feasible for use in large population-based studies in adults and elderlies. The OPEN application—connected with national food composition data, and the national-specific food portions picture book, were found very useful for efficient conduction of the study, supporting conduction of future studies. Efficient and regular monitoring of intakes of foods and nutrients is very important due to quick lifestyle changes and shifts in the availability and composition of processed foods; such studies also enable verification of the efficiency of policy interventions.

## Figures and Tables

**Table 1 nutrients-14-03618-t001:** Characteristics of adult and elderly participants in the Slovenian national dietary survey SI.Menu 2017/18 (N = 780).

		Adults (N = 364)	Elderly (N = 416)
		(18–64 Years)	(65–74 Years)
Age: years, mean (SD *)		43.6 (13.8)	68.7 (2.7)
Sex, n (%)	Female	191 (52.5)	203 (48.8)
	Male	173 (47.5)	213 (51.2)
Place of living, n (%)	Rural	202 (55.5)	229 (55.1)
	Semi-urban	56 (15.4)	71 (17.1)
	Urban	106 (29.1)	116 (27.9)
Education, n (%)	No university degree	249 (68.4)	342 (82.2)
	University degree	115 (31.6)	74 (17.8)
Self-reported socio-economic standard, n (%)	Below average	118 (38.4)	269 (71.5)
	Above average	189 (61.6)	107 (28.5)
BMI ^#^, mean (SD *)		26.7 (5.2)	28.4 (5.0)
n (%)	Underweight	4 (1.1)	5 (1.2)
	Normal	144 (39.6)	103 (24.8)
	Overweight or obese	216 (59.3)	308 (74.0)
IPAQ **, n (%)	Low intensity	127 (35.3)	137 (33.4)
	Moderate	108 (30.0)	133 (32.4)
	High intensity	125 (34.7)	140 (34.2)
Employment status, n (%)	Employed	226 (62.1)	n.a.
	Unemployed	42 (11.5)	n.a.
	Student	32 (8.8)	n.a.
	Retired	64 (17.6)	n.a.
Cohesion regions of Slovenia ^##^, n (%)	East	225 (61.8)	239 (57.5)
	West	139 (38.2)	177 (42.5)

* SD—standard deviation. ^#^ BMI—body mass index. ** IPAQ—International Physical Activity Questionnaire score rank. ^##^ In line with the EU methodology Slovenia is divided into two cohesion regions, namely the cohesion region Zahodna Slovenija (west Slovenia) and the cohesion region Vzhodna Slovenija (east Slovenia).

**Table 2 nutrients-14-03618-t002:** Descriptive statistics of usual daily intakes of food groups among male and female adults (18–64 years).

	Male								Female						
Food (Sub)Groups (g(mL)/day) ^#^	True Non-Consumers N (%) ^$^	Mean	Standard Deviation	Percentile 05	Percentile 25	Median	Percentile 75	Percentile 95	Mean	Standard Deviation	Percentile 05	Percentile 25	Median	Percentile 75	Percentile 95
**Milk**	49 (13.5)	80.0	102.2	0.0	0.0	47.5	132.7	273.7	83.8	87.0	0.0	0.0	57.1	151.5	233.4
**Dairy products** (yogurt, cheese, milk cream)	7 (1.9)	107.3	128.7	0.0	19.8	54.0	180.3	361.5	127.9	127.6	0.0	30.6	80.0	220.3	367.6
Cheese	20 (5.5)	34.7	34.3	0.0	0.0	33.0	54.0	96.2	32.0	33.1	0.0	0.0	30.6	55.7	89.0
**Vegetables** (fresh and preserved/canned)	0.0	163.0	64.8	75.9	119.1	157.5	195.4	284.4	158.3	77.1	56.4	105.3	148.4	191.0	288.6
Fresh vegetables	/	121.8	53.6	46.4	82.2	112.8	156.0	225.4	124.7	66.2	49.2	81.2	114.8	152.3	253.4
Preserved and canned vegetables	/	38.7	39.2	0.0	0.0	31.6	60.6	109.6	30.2	33.0	0.0	0.0	25.9	45.8	83.5
**Fruits** (fresh, canned, dry)	0.0	162.2	142.5	0.0	0.0	154.6	260.3	414.4	226.0	151.3	0.0	122.4	214.3	320.1	521.3
Fresh fruits	/	141.1	129.6	0.0	0.0	149.7	237.3	366.2	197.9	136.0	0.0	109.4	195.3	280.4	420.6
Other fruits (canned, dry)	/	20.2	60.6	0.0	0.0	0.0	0.0	206.6	21.8	57.7	0.0	0.0	0.0	0.0	160.1
**Legumes and legumes products** (kidney beans, green beans, lentils, etc.)	18 (4.9)	14.6	32.0	0.0	0.0	0.0	0.0	77.3	11.6	23.8	0.0	0.0	0.0	0.0	69.8
**Fresh and processed nuts and seeds**	122 (33.5) **	7.7	22.0	0.0	0.0	0.0	0.0	59.7	9.5	22.6	0.0	0.0	0.0	0.0	66.5
**Potatoes**	3 (0.8)	99.0	70.7	0.0	0.0	121.0	153.0	189.0	75.7	61.2	0.0	0.0	81.8	121.5	183.1
**Bread and bakery products** (all type of breads, dough, bread-based products)	0.0	210.0	96.8	84.0	145.3	191.4	267.0	396.3	144.7	81.0	4.5	85.2	132.2	193.1	293.8
Bread	1 (0.3)	176.8	80.4	72.0	122.8	172.2	217.2	336.3	118.5	67.6	0.0	77.1	105.0	157.2	233.9
**Cereal and cereal products** (breakfast cereals, pasta, rice etc.)	0.0	97.2	77.1	0.0	34.1	80.3	140.1	247.4	94.5	74.0	0.0	32.7	84.1	130.7	213.8
Breakfast cereals	95 (26.1)	34.3	46.5	0.0	0.0	21.4	44.3	124.8	39.2	48.8	0.0	0.0	24.1	60.8	130.7
Pasta, rice	0.0	62.9	57.8	0.0	0.0	53.9	103.7	160.2	54.7	53.6	0.0	0.0	40.6	86.8	163.6
**Fish and fish products** (fresh fish, canned fish etc.)	32 (8.8)	26.5	58.0	0.0	0.0	0.0	0.0	154.1	18.2	46.7	0.0	0.0	0.0	0.0	119.2
Fresh fish	/	13.8	50.4	0.0	0.0	0.0	0.0	154.1	9.3	34.6	0.0	0.0	0.0	0.0	106.7
**Fresh meat** (red meat and poultry)	6 (1.6)	208.6	101.8	81.8	131.6	199.2	257.6	419.1	140.8	76.2	0.0	85.8	146.1	187.2	254.8
Red meat	8 (2.2)	136.7	71.4	47.2	96.8	123.8	162.1	257.0	75.9	51.3	0.0	49.7	78.4	102.3	150.2
Poultry	10 (2.7)	71.9	72.3	0.0	0.0	69.6	131.9	198.1	64.9	55.5	0.0	0.0	77.2	109.2	145.2
**Processed meat** (sausages, salami, and other processed meat)	43 (11.8)	51.5	62.6	0.0	0.0	0.0	114.7	152.8	27.7	43.4	0.0	0.0	0.0	69.3	111.2
**Fruit and vegetable juices**	89 (24.5)	78.5	133.2	0.0	0.0	0.0	134.5	365.5	46.0	87.4	0.0	0.0	0.0	52.8	247.1
**Sugar-containing soft beverages**	114 (31.3)	136.4	253.4	0.0	0.0	0.0	250.1	704.4	62.6	177.2	0.0	0.0	0.0	0.0	393.9
**Tap and bottled water**	0.0	924.6	508.4	284.2	555.2	914.3	1213.9	1851.3	969.9	530.4	253.2	598.7	877.7	1238.2	1900.9
**Hot drinks** (coffee, tea, cacao and hot chocolate, etc.)	4 (1.1)	243.7	207.2	0.0	99.1	206.8	365.8	620.7	323.1	267.1	0.0	130.5	253.6	464.9	878.5
Coffee	55 (15.1)	91.4	108.7	0.0	0.0	69.7	148.4	285.5	104.3	112.9	0.0	0.0	108.0	155.5	286.9
Tea	23 (6.3)	106.1	177.6	0.0	0.0	0.0	243.5	466.1	172.0	228.3	0.0	0.0	0.0	274.2	627.6
**Fats and oils** (vegetable oils, margarines, butter, and other animal fat)	0.0	28.2	20.5	8.0	15.1	21.5	38.2	69.6	22.6	14.0	7.1	13.0	18.8	29.6	50.9
Vegetable oils and margarines	0.0	19.8	15.4	6.9	12.3	16.3	21.2	43.0	15.7	9.3	5.4	9.7	14.0	18.5	34.9
Butter and other animal fat	28 (7.7)	8.4	13.5	0.0	0.0	0.0	14.5	38.7	6.9	10.8	0.0	0.0	0.0	11.6	27.7
**High sugar food** (sugar, confectionary, cakes, cookies, desserts)	1 (0.3)	98.2	121.1	0.0	3.8	51.6	150.4	345.2	106.5	104.6	0.0	12.9	84.9	169.4	290.6
Sugar and confectionary	5 (1.4)	24.1	32.8	0.0	0.0	11.0	39.9	89.6	18.3	22.1	0.0	0.0	10.8	32.1	65.4
Cakes, cookies	33 (9.1)	27.7	53.7	0.0	0.0	0.0	39.9	159.8	22.5	42.0	0.0	0.0	0.0	39.4	119.2
Desserts	16 (4.4)	46.4	94.0	0.0	0.0	0.0	82.2	206.0	65.7	93.7	0.0	0.0	0.0	118.3	219.4
**Fresh and food incorporated eggs** *	150 (41.2)	43.5	46.5	0.0	0.0	35.1	79.1	129.1	36.0	33.9	0.0	0.0	33.1	62.5	92.6
**Alcoholic beverages** (wine, beer, spirits)	62 (17.0)	233.4	336.9	0.0	0.0	30.1	484.6	916.4	65.6	159.9	0.0	0.0	0.0	28.5	493.3
Wine	96 (26.4)	32.8	62.8	0.0	0.0	0.0	31.6	185.4	17.8	44.3	0.0	0.0	0.0	0.0	130.5
Beer	115 (31.6)	183.4	314.9	0.0	0.0	0.0	372.7	913.1	44.0	145.6	0.0	0.0	0.0	0.0	493.3
Spirits	338 (92.9)	17.3	73.9	0.0	0.0	0.0	0.0	112.4	3.8	23.4	0.0	0.0	0.0	0.0	0.0
**Ready to eat meals**	126 (34.6)	29.2	50.5	0.0	0.0	0.0	48.4	153.0	27.0	50.4	0.0	0.0	0.0	43.4	135.7

* Habitual intake of eggs calculated from 2 × 24HDR. ** Data for fresh nuts and seeds. ^$^ Number of consumers (both sexes) of a given food group identified in a probabilistic model by zero frequency of consumption as well as no reported consumed quantities. ^#^ Subcategories ‘Other’ not presented.

**Table 3 nutrients-14-03618-t003:** Descriptive statistics of usual daily intakes of food groups among male and female elderlies (65–74 years).

	Male								Female						
Food (Sub)Groups (g(mL)/day) ^#^	True Non-Consumers N (%) ^$^	Mean	Standard Deviation	Percentile 05	Percentile 25	Median	Percentile 75	Percentile 95	Mean	Standard Deviation	Percentile 05	Percentile 25	Median	Percentile 75	Percentile 95
**Milk**	52 (12.5)	59.9	83.6	0.0	0.0	31.7	91.6	229.7	76.3	80.8	0.0	0.0	58.4	121.3	229.7
**Dairy products** (yogurt, cheese, milk cream)	6 (1.4)	91.2	102.5	0.0	0.0	52.6	179.7	283.4	129.5	123.5	0.0	20.9	99.8	219.3	337.9
Cheese	31 (7.5)	27.4	30.6	0.0	0.0	23.5	53.8	82.6	24.7	28.4	0.0	0.0	0.0	48.2	80.1
**Vegetables** (fresh and preserved/canned)	0.0	160.1	86.2	64.0	101.0	145.9	196.0	345.2	163.2	74.2	66.1	111.2	150.2	214.6	296.0
Fresh vegetables	/	120.6	70.2	40.8	74.9	101.0	149.2	275.7	128.6	67.7	47.9	77.5	113.9	169.1	259.3
Preserved and canned vegetables	/	38.5	43.7	0.0	0.0	24.5	66.0	125.3	29.7	39.1	0.0	0.0	22.2	45.5	122.2
**Fruits** (fresh, canned, dry)	0.0	210.7	163.8	0.0	40.4	218.8	320.0	494.7	267.8	155.8	0.0	153.3	251.3	377.4	505.3
Fresh fruits	/	178.2	143.1	0.0	0.0	179.2	283.4	408.0	221.9	124.2	0.0	152.0	208.6	289.8	444.5
Other fruits (canned, dry)	/	30.4	82.5	0.0	0.0	0.0	0.0	230.2	38.8	87.4	0.0	0.0	0.0	0.0	251.3
**Legumes and legumes products** (kidney beans, green beans, lentils, etc.)	5 (1.2)	21.3	40.3	0.0	0.0	0.0	30.9	114.0	17.4	30.7	0.0	0.0	0.0	35.5	77.1
**Fresh and processed nuts and seeds**	205 (49.3) **	4.5	14.7	0.0	0.0	0.0	0.0	52.8	6.8	18.6	0.0	0.0	0.0	0.0	54.1
**Potatoes**	3 (0.7)	99.0	64.3	0.0	61.7	117.7	143.9	184.6	87.8	50.8	0.0	64.5	102.0	124.2	155.1
**Bread and bakery products** (all types of breads, dough, bread-based products)	1 (0.2)	188.2	81.1	75.9	131.5	182.5	247.0	333.0	161.4	73.9	69.0	115.1	156.2	205.9	292.3
Bread	1 (0.2)	178.1	76.1	71.4	125.7	169.7	220.0	319.7	146.2	68.1	58.2	104.9	142.3	182.8	247.3
**Cereal and cereal products** (breakfast cereals, pasta, rice, etc.)	1 (0.2)	85.9	60.9	0.0	39.3	73.6	121.7	210.9	69.9	52.1	0.0	30.0	60.3	99.7	163.0
Breakfast cereals	181 (43.5)	28.9	31.8	0.0	0.0	22.0	36.2	94.1	33.0	35.2	0.0	0.0	23.3	47.7	99.4
Pasta, rice	4 (1.0)	20.2	33.7	0.0	0.0	0.0	44.6	88.3	10.4	22.4	0.0	0.0	0.0	0.0	61.4
**Fish and fish products** (fresh fish, canned fish, etc.)	37 (8.9)	19.8	47.9	0.0	0.0	0.0	0.0	113.7	21.2	50.1	0.0	0.0	0.0	0.0	148.4
Fresh fish	/	7.9	38.8	0.0	0.0	0.0	0.0	0.0	14.6	46.8	0.0	0.0	0.0	0.0	148.4
**Fresh meat** (red meat and poultry)	1 (0.2)	188.7	88.6	75.3	123.9	177.5	231.2	341.5	141.1	80.9	0.0	82.6	131.3	181.0	273.5
Red meat	2 (0.5)	132.9	69.5	31.9	92.3	123.9	164.6	252.6	93.5	65.7	0.0	59.2	83.4	122.7	200.6
Poultry	7 (1.7)	55.8	63.4	0.0	0.0	0.0	109.2	170.1	47.6	51.4	0.0	0.0	50.8	86.4	128.6
**Processed meat** (sausages, salami, and other processed meat)	38 (9.1)	41.4	54.3	0.0	0.0	0.0	97.6	134.5	24.9	43.0	0.0	0.0	0.0	60.6	107.2
**Fruit and vegetable juices**	151 (36.3)	42.9	104.5	0.0	0.0	0.0	42.0	273.1	42.0	82.0	0.0	0.0	0.0	39.9	229.5
**Sugar-containing soft beverages**	227 (54.6)	40.7	128.7	0.0	0.0	0.0	0.0	421.8	24.0	91.9	0.0	0.0	0.0	0.0	249.6
**Tap and bottled water**	0.0	730.4	462.5	140.0	400.0	650.0	927.9	1627.3	790.8	429.9	216.1	423.8	733.8	1101.7	1580.7
**Hot drinks** (Coffee, tea, cacao and hot chocolate, etc.)	3 (0.7)	320.0	239.5	0.0	152.2	266.3	423.3	829.3	342.1	216.4	81.7	164.9	354.4	454.9	769.3
Coffee	43 (10.3)	88.2	83.7	0.0	0.0	106.8	135.6	238.1	103.7	92.0	0.0	0.0	108.6	154.1	199.5
Tea	25 (6.0)	184.1	228.5	0.0	0.0	60.4	317.7	564.9	171.1	197.3	0.0	0.0	150.5	266.4	571.8
**Fats and oils** (Vegetable oils, margarines, butter, and other animal fat)	0.0	26.8	20.7	7.8	13.6	20.5	31.8	74.1	26.4	14.9	7.5	14.9	24.3	35.2	54.8
Vegetable oils and margarines	1 (0.2)	16.4	10.5	6.0	9.6	14.3	19.3	36.3	15.6	9.7	4.8	9.8	13.0	18.2	33.5
Butter and other animal fat	19 (4.6)	10.5	16.1	0.0	0.0	0.0	15.8	44.8	10.9	12.7	0.0	0.0	7.5	17.7	35.3
**High sugar food** (sugar, confectionary, cakes, cookies, desserts)	2 (0.5)	84.2	106.0	0.0	7.7	40.8	128.5	336.2	90.0	91.6	0.0	16.2	57.0	144.8	270.0
Sugar and confectionary	5 (1.2)	20.3	27.7	0.0	0.0	10.9	30.7	77.3	25.3	26.9	0.0	4.2	18.5	41.0	71.6
Cakes, cookies	51 (12.3)	22.1	44.1	0.0	0.0	0.0	38.5	132.3	20.1	43.7	0.0	0.0	0.0	24.8	105.7
Desserts	23 (5.5)	41.8	90.3	0.0	0.0	0.0	0.0	246.1	44.7	72.7	0.0	0.0	0.0	97.0	183.7
**Fresh and food incorporated eggs** *	183 (44.0)	41.1	42.1	0.0	0.0	35.1	73.5	122.3	29.3	31.5	0.0	0.0	31.6	52.1	88.4
**Alcoholic beverages** (wine, beer, spirits)	79 (19.0)	220.2	275.6	0.0	0.0	106.6	364.2	761.6	37.1	79.9	0.0	0.0	0.0	30.7	189.4
Wine	113 (27.2)	78.5	135.6	0.0	0.0	0.0	126.0	344.9	21.9	50.0	0.0	0.0	0.0	0.0	122.4
Beer	183 (44.0)	112.9	234.2	0.0	0.0	0.0	0.0	532.6	8.4	48.4	0.0	0.0	0.0	0.0	0.0
Spirits	364 (87.5)	28.8	75.6	0.0	0.0	0.0	0.0	204.0	6.8	30.9	0.0	0.0	0.0	0.0	20.5
**Ready to eat meals**	215 (51.7)	26.3	49.9	0.0	0.0	0.0	27.7	167.7	28.0	58.4	0.0	0.0	0.0	19.7	169.8

* Habitual intake of eggs calculated from 2 × 24HDR. ** Data for fresh nuts and seeds. ^$^ Number of consumers (both sexes) of a given food group identified in a probabilistic model by zero frequency of consumption as well as no reported consumed quantities. ^#^ Subcategories ‘Other’ not presented.

**Table 4 nutrients-14-03618-t004:** Descriptive statistics of usual daily intakes of energy, macronutrients, dietary fibre, and water among male and female adults (18–64 years).

	Male								Female							
	Mean	Standard Deviation	Standard Error of Mean	Percentile 05	Percentile 25	Median	Percentile 75	Percentile 95	Mean	Standard Deviation	Standard Error of Mean	Percentile 05	Percentile 25	Median	Percentile 75	Percentile 95
**Energy** (kJ/day)	9587	2663	202	5618	7580	9417	11,445	14,040	7356	1749	127	4746	6006	7236	8580	10,641
kJ/kg body weight	115	36.9	2.8	58.9	84.5	111.2	143.1	171.3	107.7	32.2	2.3	60.1	82.4	105.9	129.48	169.9
**Carbohydrates** (g/day)	238	73.3	5.6	126.1	180.8	233.6	290.5	365.2	198.3	52.8	3.8	116.1	159.0	197.0	232.9	289.6
kJ/day	3980	1227	93	2112	3029	3913	4865	6116	3322	884	64	1945	2663	3299	3900	4850.1
% TEI	41.9	8.2	0.6	27.5	36.7	42.0	47.1	55.1	45.6	8.4	0.6	30.5	40.5	46.1	51.3	59.0
**Total Sugars** ^a^ (g/day)	68.3	38.1	2.9	13.7	40.1	66.2	91.9	138.5	65.4	26.8	1.9	29.1	47.6	61.5	77.7	110.4
kJ/day	1196	668	51	239	702	1159	1608	2424	1145	469	34	509	833	1077	1360	1932
% TEI	12.3	5.6	0.4	3.1	8.6	12.2	15.3	22.7	15.8	6.0	0.4	7.9	11.9	15.0	19.4	25.9
**Free sugars** (g/day)	38.7	33.7	2. 6	1.1	12.9	30.8	57.9	103.1	29.8	22.9	1.7	2.0	14.8	24.0	41.0	70.6
kJ/day	678	590	45	19	225	539	1013	1805	522	401	29	36	258	421	718	1235
% TEI	6.7	5.4	0.4	0.3	2.9	5.5	9.6	17.1	7.1	5.1	0.4	0.5	3.8	6.1	9.6	17.1
**Proteins** (g/day)	114	37.7	2.9	60.3	86.7	110.7	133.3	190.8	81.9	23.9	1.7	42.2	66.7	80.3	96.2	123.8
kJ/day	1916	631	48	1010	1452	1854	2233	3196	1371	400	29	707	1117	1345	1611	2073
% TEI	20.1	3.7	0.3	15.2	17.6	19.9	22.2	27.0	18.7	3.5	0.3	13.3	16.5	18.6	20.5	25.5
g/kg body weight	1.4	0.5	0.0	0.7	1.0	1.4	1.7	2.04	1.2	0.4	0.0	0.6	0.9	1.20	1.5	1.96
**Total Fats** (g/day)	87.5	30.4	2.3	43.2	66.9	85.7	109.4	143.6	67.6	23.6	1.7	33.1	49.9	65.9	83.9	114.9
kJ/day	2563	890	68	1266	1961	2511	3207	4208	1982	691	50	971	1463	1930	2460	3368
% TEI	26.6	5.5	0.4	17.8	22.7	26.5	30.6	35.3	26.6	5.6	0.4	18.0	22.7	26.5	29.9	37.2
**Dietary Fibres** (g/day)	21.4	7.5	0.6	10.4	16.3	20.3	25.3	35.3	20.3	6.6	0.5	10.5	15.7	19.0	23.4	33.4
g/MJ	2.3	0.9	0.1	1.2	1.72	2.17	2.7	3.9	2.8	0.9	0.1	1.7	2.2	2.7	3.2	4.2
**Total water** ^b^ (mL/day)	2438	691	53	1392	2006	2379	2888	3628	2260	649	47	1225	1834	2184	2592	3450

^a^ Total sugars: free sugars and sugars naturally occurring in foods (e.g., lactose in milk, fructose in fruits) [35]. ^b^ Total water refers to the total water intake from beverages and solid foods.

**Table 5 nutrients-14-03618-t005:** Descriptive statistics of usual daily intakes of energy, macronutrients, and dietary fibre among elderly aged 65–74, by gender.

	Male								Female							
	Mean	Standard Deviation	Standard Error of Mean	Percentile 05	Percentile 25	Median	Percentile 75	Percentile 95	Mean	Standard Deviation	Standard Error of Mean	Percentile 05	Percentile 25	Median	Percentile 75	Percentile 95
**Energy** (kJ/day)	8638	2215	152	5467	7071	8350	10,072	12,731	7112	1668	117	4547	5954	6973	8089	10,284
kJ/kg body weight	102.9	30.9	2.12	60.4	79.5	99.3	123.0	162.3	100.2	30.7	2.1	58.2	80.4	95.9	117.2	164.0
**Carbohydrates** (g/day)	212.9	61.8	4.2	116.4	170.3	210.1	246.2	326.0	196.7	50.8	3. 6	116.0	162.3	195.3	229.2	286.9
kJ/day	3566	1035	70.9	1950	2853	3519	4124	5459	3295	850	59.7	1943	2719	3271	3839	4805
% TEI	41.8	8.8	0.6	28.1	35.6	42.0	46.8	56.8	46.8	8.6	0.6	32.5	40.7	47.0	52.5	61.1
**Total Sugars** ^a^ (g/day)	59.4	30.7	2.1	17.4	32.9	57.1	77.3	113.2	65.2	25.4	1.8	29.5	45.6	61.8	80.6	108.0
kJ/day	1039	538	37	304	577	1000	1354	1981	1142	445	31	517	798	1082	1411	1890
% TEI	12.2	5.8	0.4	3.9	7.6	11.9	15.5	22.9	16.3	5.8	0.4	8.5	11.6	15.5	19.8	26.6
**Free sugars** (g/day)	27.0	21.5	1.5	1.3	12.2	23.0	38.7	66.0	27.7	20.1	1.4	0.9	14.4	23.6	36.5	64.3
kJ/day	473	376	26	22	213	403	677	1155	484	352	25	16	252	414	639	1126
% TEI	5.5	4.1	0.3	0.3	2.6	4.8	7.7	12.6	6.7	4.7	0.3	0.2	3.7	5.8	9.1	15.4
**Proteins** (g/day)	105.3	34.3	2.4	61.3	83.9	100.5	121.3	169.2	80.6	23.9	1.7	46.6	64.6	76.4	105.3	34.3
kJ/day	1764	574	39	1026	1405	1683	2031	2834	1351	400	28	781	1082	1280	1764	574
% TEI	20.6	4.4	0.3	14.9	17.5	20.0	22.9	27.5	19.0	3.4	0.2	13.6	16.9	18.5	20.6	4.4
g/kg body weight	1.3	0.4	0.0	0.7	1.0	1.2	1.5	2.1	1.1	0.4	0.0	0.6	0.9	1.1	1.3	0.4
**Total Fats** (g/day)	76.5	28.8	1.97	34.1	57.6	71.2	94.9	130.7	63.2	22.2	1.6	30.1	47.9	62.9	76.5	28.8
kj/day	2242	844	58	1000	1688	2087	2782	3829	1852	652	46	881	1405	1843	2242	844
% TEI	25.7	5.9	0.4	15.6	21.5	25.8	30.1	35.1	25.7	5.7	0.4	16.6	22.0	25.7	25.7	35.9
**Dietary Fibres** (g/day)	21.5	6.9	0.5	12.2	16.6	21.2	25.8	33.2	21.6	7.0	0.5	12.0	16.5	20.6	25.8	34.8
g/MJ	2.5	0.8	0.1	1.5	2.0	2.5	2.9	3.8	3.1	1.0	0.1	1.9	2.4	3.0	3.6	4.6
**Total water** ^b^ (mL/day)	2147	588	40	1262	1693	2150	2551	3221	2084	488	34	1339	1732	2067	2430	2880

^a^ Total sugars: free sugars and sugars naturally occurring in foods (e.g., lactose in milk, fructose in fruits) [35]. ^b^ Total water refers to the total water intake from beverages and solid foods.

**Table 6 nutrients-14-03618-t006:** Proportion (%) of the study population meeting reference values for intake of specific food groups, nutrients, dietary fibre, and water.

	Adults								Elderly								Dietary Reference Values	
	Male				Female				Male				Female					
Food Groups	% Meet DRI	% Do Not Meet DRI	% Below Min DRI	% Above Max DRI	% Meet DRI	% Do Not Meet DRI	% Below Min DRI	% Above Max DRI	% Meet DRI	% Do Not meet DRI	% Below Min DRI	% Above Max DRI	% Meet DRI	% Do Not Meet DRI	% Below Min DRI	% Above Max DRI	Adults	Elderly
Milk and dairy products ^1^	14.2	85.8	71.0	14.8	17.2	82.8	68.3	14.4	11.3	88.7	78.9	9.8	9.6	90.4	72.2	18.2	250–310 g/day	
Vegetables (fresh)	29.5	70.5	-	-	25.7	74.3	-	-	24.4	75.6	-	-	30.5	69.5	-	-	>150 g/day	
Vegetables (fresh and preserved/canned)	3.5	96.5	-	-	6.8	93.2	-	-	9.4	90.6	-	-	8.9	91.1	-	-	>250 g/day	
Fruits	33.1	66.9	-	-	43.0	57.0	-	-	48.4	51.6	-	-	50.0	50.0	-	-	>250 g/day	
Potatoes, pasta, rice	6.2	93.8	92.0	1.9	2.9	97.1	96.0	1.1	4.0	96.0	94.5	1.5	1.0	99.0	99.0	0.0	200–250 g/day	
Bread and cereal products	42.4	57.6	48.8	8.7	19.6	80.4	79.9	0.5	37.6	62.4	58.2	4.2	15.8	84.2	82.7	1.5	200–310 g/day	
Fish and fish products ^2^	26.4	73.6	29.8	43.8	24.8	75.2	26.7	48.5	22.6	77.4	17.6	59.7	22.3	77.7	20.2	57.4	22–32 g/day	
Meat (fresh and processed) ^2^	1.2	98.8	1.3	97.7	16.5	83.5	1.6	81.9	2.8	97.2	0.0	97.2	17.4	82.6	0.5	82.1	43–86 g/day	
Eggs ^2^	99.0	1.0	-	-	100.0	0.0	-	-	97.6	2.4	-	-	100.0	0.0	-	-	<26 g/day	
Added fats	25.0	75.0	72.7	2.3	15.3	84.7	84.7	0.0	14.1	85.9	82.2	3.8	16.3	83.7	82.7	1.0	25–45 g/day	
Water and non-alcoholic beverages ^3^	21.4	78.6	78.6	0.0	24.1	75.9	75.9	0.0	11.8	88.2	88.2	0.0	12.3	87.7	-	-	>1500 mL/day	
**Macronutrients**																		
Carbohydrates	14.5	85.5	-	-	29.8	70.2	-	-	16.0	84.0	-	-	36.5	63.5	-	-	>50% of total energy intake	
Free sugars (criteria 1) ^b^	77.5	22.5	-	-	77.0	23.0	-	-	88.7	11.3	-	-	82.8	17.8	-	-	<10% of total energy intake	
Free sugars (criteria 2) ^b^	43.9	56.5	-	-	38.7	61.3	-	-	51.2	48.8	-	-	40.4	59.6	-	-	<5% of total energy intake	
Proteins (criteria 1)	97.1	2.9	-	-	91.6	8.4	-	-	92.5	7.5	-	-	86.7	13.3	-	-	>48/57 g/day ^#^	>57/67 g/day ^#^
Proteins (criteria 2)	89.0	11.0	-	-	82.2	17.8	-	-	70.9	21.1	-	-	56.7	43.3	-	-	>0.8 g/kg of b. w.	>1.0 g/kg of b. w
Total Fats	71.7	28.3	-	-	75.9	24.1	-	-	74.2	25.8	-	-	79.3	20.7	-	-	<% 30 of total energy intake	
Dietary Fibres	13.3	86.7	-	-	8.4	91.6	-	-	10.8	89.2	-	-	13.3	86.7	-	-	>30 g/day	
Total water ^a^	41.6	58.4	-	-	62.8	37.2	-	-	27.7	72.3	-	-	56.7	48.3	-	-	>2000/2500 mL/day ^#^	>2000/2500 mL/day ^#^

Recommendations for food groups according to DGE Nutrition Circle [25]. For ranges, the proportion of the study population with excess/insufficient consumption is reported. Recommendations for nutrients and dietary fibre according to D-A-CH nutrient reference values (DRV) [33]. Recommendations for total water according to adequate intake in EFSA’s dietary reference values for water [34]. ^1^ Sum of recommendations for ‘milk and dairy products’ and ‘cheese’. ^2^ Weekly recommendations converted to a daily value. ^3^ Water and non-alcoholic beverages refer bottled and tap water, coffee, tea, and all types of non-alcoholic beverages. ^a^ Total water intake refers to the total water from beverages and solid foods. ^b^ WHO guidelines for free sugar intake [35]. ^#^ Sex-dependent recommendations (females/males).

## Data Availability

Food consumption data presented in this study are available on EFSA web-page: https://www.efsa.europa.eu/en/data-report/food-consumption-data (accessed on 28 January 2022), other and more detailed data on request from the NIJZ e-mail: EUmenu.SLO@nijz.si.

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
