# Peer review of "Dietary Intakes of Slovenian Adults and Elderly: Design and Results of the National Dietary Study SI.Menu 2017/18"

_nutrients, 2022, doi:10.3390/nu14173618_

Round 1
Reviewer 1 Report
This is a very interesting paper in that it reveals the current status and challenges of nutritional intake in Slovenia through a large-scale survey. The test itself with such a large subject is valuable. I think this manuscript is worth to publish in “Nutrients”. But some additional modification should be needed.
1) Line 46-47: Sufficient references showing relationship between diets and cardiovascular disease, cancer, and diabetes should be shown.
2) This manuscript mentioned that the average of BMI was high, but it is important to understand how much it deviates from the national average. This should be noted in the results or discussion. Also, the university enrollment rate in Slovenia needs to be cited. What is the reason for the bias in the subject?
3) For fruits, there is a wide range, from polyphenol-rich fruits to those that do not contribute much to health; what fruits were commonly eaten in the Adult or elder should be noted in the results.
4) The oils ranged from those high in saturated fatty acids, such as rapeseed oil, to those high in unsaturated fatty acids, such as grape seed oil. Information about what kind of oil the subject group preferred to consume, especially unsaturated fatty acids, is very important and should be considered for additional analysis.
5) Beer consumption is mentioned in the results, but Slovenia has the image of being a wine consuming country. Since the effects of polyphenols in wine are well known, it would be desirable to include information on wine as well.
6) Very shocking results were reported that the nutritional intake status of many people deviates significantly from the amounts recommended in the guidelines. Possible measures to solve this problem should be described in more detail in the discussion section.
Reviewer 2 Report
This experiment involved a total of 780 people and generated a huge amount of data, which required a lot of time and effort. The data obtained are also very representative. However, I noticed that the Adults group with the age ranging from 18 to 64 years old, with a gap of 46 years, had only 364 people; while the Elderly group with the age ranging from 65 to 74 years old, with a gap of 9 years, had 416 people. I think the eating habits of a 64-year-old and a 65-year-old should be very similar. Such grouping may not be appropriate enough. It might be better if you could group by every 10 years.
Meanwhile, the authors noted the differences between urban and rural populations in consumption and eating habits, but they were not discussed separately in the article.
As mentioned in the text about the BMI, when it is lower than 18.5, it should belong to underweight. But there were only Normal and Overweight in Table 1. Is there some data missing?
I noticed that many classifications were not accurate enough. Such as meat and fruit. The nutrients provided by different fruits are completely different. Body weight is also a factor that significantly affects food intake. Has the author considered choosing a more precise way for the evaluation, such as meat (fruit) intake(g)/unit body weight(kg)?
Has the author considered the differences in income that affect people's purchasing power?
Round 2
Reviewer 1 Report
I have confirmed that the points I have pointed out have been sufficiently corrected.